# THE VARIATIONAL WALKBACK ALGORITHM

**Anirudh Goyal**[*], **Nan Rosemary Ke**[†], **Alex Lamb**[‡], **Yoshua Bengio**[§]

## ABSTRACT

A recognized obstacle to training undirected graphical models with latent variables such as Boltzmann machines is that the maximum likelihood training procedure requires sampling from Monte-Carlo Markov chains which may not mix well, in the inner loop of training, for each example. We first propose the idea that it is sufficient to locally carve the energy function everywhere so that its gradient points in the "right" direction (i.e., towards generating the data). Following on previous work on contrastive divergence, denoising autoencoders, generative stochastic networks and unsupervised learning using non-equilibrium dynamics, we propose a variational bound on the marginal log-likelihood of the data which corresponds to a new learning procedure that first walks away from data points by following the model transition operator and then trains that operator to walk backwards for each of these steps, back towards the training example. The tightness of the variational bound relies on gradually increasing temperature as we walk away from the data, at each step providing a gradient on the parameters to maximize the probability that the transition operator returns to its previous state. Interestingly, this algorithm admits a variant where there is no explicit energy function, i.e., the parameters are used to directly define the transition operator. This also eliminates the explicit need for symmetric weights which previous Boltzmann machine or Hopfield net models require, and which makes these models less biologically plausible.

## 1 INTRODUCTION

Although earlier research focused on generating data through Monte Carlo Markov chains (MCMCs), e.g. with various Boltzmann machines (Salakhutdinov & Hinton, 2009), most of the recent effort in designing deep generative models is based on single-step generation, e.g., with variational auto-encoders (VAEs) (Kingma & Welling, 2013; Rezende et al., 2014) and generative adversarial networks (GANs) (Goodfellow et al., 2014). However, generating a sample by going through a series of stochastic transformations that gradually improve the generated sample (or its latent representation) to make it more plausible could hold some advantages. A generative process can be seen as a mapping from simple noise variates (e.g., uniform, Gaussian) to samples from a very complicated distribution (maybe concentrated near a low-dimensional manifold) approximating the one which we are trying to learn from. If the data distribution is complex (e.g., the corresponding manifold is highly convoluted and non-linear), the generative process may involve a highly non-linear transformation which could be difficult to learn and optimize. Such highly non-linear transformations are probably best represented (and learned) by composing a large number of slightly non-linear transformations, either with a fixed-depth deep network, or with a variable depth recurrent computation, which is what the repeated application of a transition operator corresponds to.

### 1.1 MOTIVATIONS

The main motivation for the paper are the following.

- The main difference between feedforward generation and recurrent generation is two-fold:(1) in the recurrent case, the same parameters are used for each step of the transition

---

[*]anirudhgoyal9119@gmail.com

[†]rosemary.nan.ke@gmail.com

[‡]lambalex@iro.umontreal.ca

[§]CIFAR Senior Fellow

operator, and (2) by providing an interpretation of each of these steps as the application of a transition operator, we can design training procedures which do not require back-propagating through all the steps of the unfolded computation (from the raw noise samples to the generated output). This is a potential that clearly deserves to be explored further and motivates the learning framework introduced here.

- Another motivation for the Variational Walkback is the idea that we only need to carve the energy function in the right direction at each point in the space of the random variables of interest, which may sideskip the need to actually sample from the stationary distribution of a Markov chain in order to obtain the gradients of the training objective. The intuition is that if the model's transition operator wants to move away from the data and into an area without data, this is a clue that the energy gradient is pointing in the wrong direction at that place. Consider a chain of samples following the model's transition operator (or variants of it at different temperatures), starting at a data point. If the chain moves us away from data points, then we can use the previous state in the chain as a target for the operator when that operator is applied to the next next state, i.e., we want to teach the operator to *walk back* towards the data. This intuition was already exploited by Bengio et al. (2013c) but without a firm mathematical grounding. In Variational Walkback this is rigorously justified by a variational bound.

- Yet another motivation for the particular approach presented here is that it innovates in the rarely explored direction of parametrizing directly the generative model via a transition operator, rather than via an explicit probability function or energy function. This idea has already been discussed in the context Generative Stochastic Networks (GSNs) (Bengio et al., 2013b), a generalization of denoising auto-encoders (DAEs) (Vincent et al., 2008) which interprets the auto-encoder as estimating the gradient of an energy function (Alain & Bengio, 2014) or as a transition operator (Bengio et al., 2013c). An advantage of being able to parametrize directly the generator is seen with GANs and DAEs: we directly parametrize and learn the function which will be used to perform the task of interest (e.g. generating answers to some questions). Instead, the traditional approach is to parametrize a probability function or energy function (e.g., with a Boltzmann machine) and then then use another procedure (the MCMC method of your choice) to sample from it and do inference. Another important reason for exploring algorithms for directly learning a transition operator is that they put less constraint on the form of the transition operator, compared with a transition operator derived from an energy function. More specifically, neural net implementations of transition operators derived from an MCMC typically require the presence of symmetric weights (due to the symmetry of the second derivative of the energy with respect to a pair of units in the neural network), as discussed by Bengio et al. (2015). When we consider a biologically plausible implementation of these learning algorithms, the weight symmetry constraint ($W_{ij} = W_{ji}$) is not reasonable as a hard constraint. Instead, if the transition operator (rather than the energy function) is the object being parametrized and learned, then there is no such hard constraint.

## 1.2 GENERAL THEORY

We introduce a novel variational bound which is an alternative to and improves upon the traditional reconstruction error as a training objective for DAEs and GSNs. Similar variational bounds have been used for VAEs as well as for the non-equilibrium thermodynamics generative models (Sohl-Dickstein et al., 2015). A distribution $P$ over a chain of samples is defined, which corresponds to iteratively applying transition operators with shared parameters, starting from a pure noise initial state. We would like this process to produce training examples. An inverting flow $Q$ is defined starting from a training example (the "walk-away" trajectory), and following the transition operator of the model, i.e., estimating the posterior distribution of the generative chain produced by $P$, given that it were landing at a training example. If the model does not match the data distribution, that chain $Q$ will tend to walk away from the training samples, and we want to inhibit that by training $P$ to "walk back". Instead of using a completely different parametrization for the variational approximation of the posterior (the $Q$ distribution), like in VAEs and non-equilibrium dynamics, we propose to exploit the decomposition of $P$ as a series of stochastic transformations in order to parametrize $Q$ with the same parameters as $P$, with the step-wise estimated posterior matching the correct one (from $P$) for all but the last step of the walk-away trajectory. To make the approximation in the

last step of the chain of walk-away steps better (and thus the variational bound tighter) we introduce the idea of gradually increasing temperature at each step of the walk-away $Q$ chain of transitions (or gradually reducing temperature, at each step of the corresponding walkback trajectory under $P$). This also has the advantage that the training procedure will more easily converge to and eliminate spurious modes (those modes of the model where there is no nearby training data). This is because the walk-away $Q$ chain will be making large steps towards the dominant and most attractive modes when the temperature becomes large enough. Unless those modes are near data points, the walkback algorithm will thus "seek and destroy" these modes, these spurious modes.

We present a series of experimental results on several datasets illustrating the soundness of the proposed approach on the MNIST, CIFAR-10 and CelebA datasets.

## 2 MIXING-FREE TRAINING FRAMEWORK BASED ON THE WALKBACK IDEA

### 2.1 MAXIMUM LIKELIHOOD TRAINING OF UNDIRECTED GRAPHICAL MODELS

Let $v$ denote the vector of visible units and $h$ denote the vector of hidden random variables, with the full state of the model being $s = (v, h)$. Let $p_\theta$ denote the model distribution, with joint energy function $E_\theta$ and parameter vector $\theta$:

$$p_\theta(s) := \frac{e^{-E_\theta(s)}}{Z_\theta},$$

(1)

where $Z_\theta$ is the partition function

$$Z_\theta := \int e^{-E_\theta(s)} ds.$$

(2)

Let $p_\mathcal{D}$ be the training distribution, from which a sample $\mathcal{D}$ is typically drawn to obtain the training set. The maximum likelihood parameter gradient is

$$\mathbb{E}_{v \sim p_\mathcal{D}} \left[ -\frac{\partial \log p_\theta(v)}{\partial \theta} \right] = \mathbb{E}_{v \sim p_\mathcal{D}, h \sim p_\theta(h|v)} \left[ \frac{\partial E_\theta(v, h)}{\partial \theta} \right] - \mathbb{E}_{s \sim p_\theta(s)} \left[ \frac{\partial E_\theta(s)}{\partial \theta} \right]$$

(3)

which is zero when training has converged, with expected energy gradients in the positive phase (under $p_\mathcal{D}(v)p_\theta(h|v)$) matching those under the negative phase (under $p_\theta(s)$). Note that in the (common) case of a log-linear model, the energy gradient (with respect to parameters) corresponds to the sufficient statistics of the model. Training thus consists in matching the shape of two distributions, as captured by the sufficient statistics: the positive phase distribution (influenced by the data, via the visible) and the negative phase distribution (where the model is free-running and generating configurations by itself).

### 2.2 MIXING-FREE TRAINING FRAMEWORK FOR UNDIRECTED GRAPHICAL MODELS

The basic idea of the proposed mixing-free training framework for undirected graphical models is the following. Instead of trying to match the whole positive phase and negative phase distributions (each of which require a difficult sampling operation, generally with an MCMC that may take very long time to mix between well separated modes), we propose to only match the shape of the energy function locally, around well-chosen points $s_t$. Another way to think about this is that instead of trying to directly maximize the likelihood of $p_\theta$ which requires expensive inference (ideally an MCMC) in the inner loop of training (for each example $v \sim p_\mathcal{D}$), we would like to learn a transition operator $p_T(s_{t+1}|s_t)$ such that following it at temperature $T = 1$ would gradually move the state $s_t$ towards the data generating distribution.

For this purpose, we propose to use a *walkback* strategy similar to the one introduced by Bengio et al. (2013c), illustrated in Algorithm 1. The idea is to start from a configuration of $s$ which is compatible with the observed data $x$, let the state evolve according to our transition operator, and then punish it for these moves, making it more likely to make backwards transitions on this trajectory. If learning was completed, the only moves that would remain are those between highly probable configurations under the data generating distribution. The other ones would be "punished",

like a child walking away from its designated task and forced to walk back (towards the data)[1]. Following the model's inclination in order to generate this random trajectory is more efficient than simply adding noise (like in the denoising auto-encoder (Vincent et al., 2008) or the non-equilibrium dynamics (Sohl-Dickstein et al., 2015) algorithms) because it makes the learning procedure focus its computation on state configurations corresponding to spurious modes to be eliminated. To make sure these spurious modes are approached efficiently, the proposed algorithm also includes the idea of gradually increasing temperature (i.e., the amount of noise) along this walk-away trajectory. At high temperature, the transition operator mixes very easily and quickly reaches the areas corresponding to large spurious modes.

Interestingly, all this comes out naturally of the variational bound presented below, rather than as something imposed in addition to the training objective.

---

**Algorithm 1 VariationalWalkback($\boldsymbol{\theta}$)**

Train a generative model associated with a transition operator $p_T(\boldsymbol{s}|\boldsymbol{s}')$ at temperature $T$ (temperature 1 for sampling from the actual model). This transition operator injects noise of variance $T\sigma^2$ at each step, where $\sigma^2$ is the noise level at temperature 1.

---

**Require:** Transition operator $p_T(\boldsymbol{s}|\boldsymbol{s}')$ from which one can both sample and compute the gradient of $\log p_T(\boldsymbol{s}|\boldsymbol{s}')$ with respect to parameters $\theta$, given $\boldsymbol{s}$ and $\boldsymbol{s}'$.
**Require:** Precomputed $\sigma_{\text{data}}^2$, the overall variance (or squared diameter) of the data.
 **repeat**
 $T_{\max} \leftarrow \frac{\sigma_{\text{data}}^2}{\sigma^2}$
 $K \leftarrow \log_2 T_{\max}$
 Sample $\boldsymbol{x} \sim$ data (or equivalently sample a minibatch to parallelize computation and process each element of the minibatch independently)
 Let $\boldsymbol{s}_0 = (\boldsymbol{x})$ and initial temperature $T = 1$, initialize $\mathcal{L} = 0$
 **for** $t = 1$ to $K$ **do**
 Sample $\boldsymbol{s}_t \sim p_T(\boldsymbol{s}|\boldsymbol{s}_{t-1})$
 Increment $\mathcal{L} \leftarrow \mathcal{L} + \log p_T(\boldsymbol{s}_{t-1}|\boldsymbol{s}_t)$
 Update parameters with log likelihood gradient $\frac{\partial \log p_T(\boldsymbol{s}_{t-1}|\boldsymbol{s}_t)}{\partial \theta}$
 Increase temperature with $T \leftarrow 2T$
 **end for**
 Increment $\mathcal{L} \leftarrow \mathcal{L} + \log p^*(\boldsymbol{s}_K)$
 **until** convergence (monitoring $\mathcal{L}$ on a validation set and doing early stopping)

---

## 3 VARIATIONAL LOWER BOUND ON THE LOG-LIKELIHOOD

Let us first consider a way in which our model could approximately generate samples according to our model and the associated transition operator $p_T(\boldsymbol{s}|\boldsymbol{s}')$. That process would start by sampling a state $\boldsymbol{s}_K$ inside a volume that contains all the data, e.g., with a broad Gaussian $p^*(\boldsymbol{s}_K)$ whose variances are set according to the training data. Then we would sample $\boldsymbol{s}_{K-1}$ from $p_{T_{\max}}(\boldsymbol{s}|\boldsymbol{s}' = \boldsymbol{s}_K)$, where $T_{\max}$ is a high enough temperature so that the noise dominates the signal and is strong enough to move the state across the whole domain of the data on the visible portion of the state. If $\sigma_{\text{data}}^2$ is the maximum variance of the data (corresponding to the visible dimensions of the state) and $\sigma^2$ is the amount noise injected by the transition operator on the visible units at temperature 1, then we could pick

$$T_{\max} = \frac{\sigma_{\text{data}}^2}{\sigma^2} \tag{4}$$

to achieve that goal. From that point on we are going to continue sampling the "previous" state $s_t$ according to $p_T(\boldsymbol{s}|\boldsymbol{s}' = \boldsymbol{s}_{t+1})$ while gradually cooling the temperature, e.g. by dividing it by 2 after each step. In that case we would need

$$K = \log_2 T_{\max} \tag{5}$$

---

[1]This analogy with a child was first used in talks by Geoff Hinton when discussing constrastive divergence (personal communication)

steps to reach a temperature of 1. Finally, we would look at the visible portion of $s_0$ to obtain the sampled $x$. In practice, we would expect that a slower annealing schedule would yield samples more in agreement with the stationary distribution of $p_1(s|s')$, but we explored this aggressive annealing schedule in order to obtain faster training.

The marginal probability of $v = x$ at the end of the above $K$-step process is thus

$$\mathrm{p}(\boldsymbol{x}) = \int_{\boldsymbol{s}_1^K} p_{T_0}(\boldsymbol{s}_0 = \boldsymbol{x}|\boldsymbol{s}_1) \left( \prod_{t=2}^K p_{T_t}(\boldsymbol{s}_{t-1}|\boldsymbol{s}_t) \right) p^*(\boldsymbol{s}_K) d\boldsymbol{s}_1^K \quad (6)$$

where $T_t$ is an annealing schedule with $T_0 = 1$ and $T_K = T_{\max}$ and $p^*$ is the "starting distribution", such as the Gaussian of variance $\sigma_{\mathrm{data}}^2$. We can rewrite this as follows by taking the log and multiplying and dividing by an arbitrary distribution $q(\boldsymbol{s}_1, \ldots, \boldsymbol{s}_K)$ decomposed into conditionals $q_{T_t}(\boldsymbol{s}_t|\boldsymbol{s}_{t-1})$:

$$\log p(\boldsymbol{x}) = \log \int_{\boldsymbol{s}_1^K} q_{T_0}(\boldsymbol{x}) q_{T_1}(\boldsymbol{s}_1|\boldsymbol{s}_0(\boldsymbol{x},)) \left( \prod_{t=2}^K q_{T_t}(\boldsymbol{s}_t|\boldsymbol{s}_{t-1}) \right)$$

$$\frac{p_{T_0}(\boldsymbol{s}_0 = \boldsymbol{x}|\boldsymbol{s}_1) \left( \prod_{t=2}^K p_{T_t}(\boldsymbol{s}_{t-1}|\boldsymbol{s}_t) \right) p^*(\boldsymbol{s}_K)}{q_{T_0}(\boldsymbol{x}) q_{T_1}(\boldsymbol{s}_1|\boldsymbol{s}_0 = \boldsymbol{x}) \left( \prod_{t=2}^K q_{T_t}(\boldsymbol{s}_t|\boldsymbol{s}_{t-1}) \right)} d\boldsymbol{s}_1^K \quad (7)$$

where we understand that $s_0 = x$. Now we can apply Jensen's inequality as usual to obtain the variational bound

$$\log p(x) \geq \mathcal{L}$$

$$= \int_{\boldsymbol{s}_1^K} q_{T_0}(\boldsymbol{x}) q_{T_1}(\boldsymbol{s}_1|\boldsymbol{s}_0 = \boldsymbol{x}) \left( \prod_{t=2}^K q_{T_t}(\boldsymbol{s}_t|\boldsymbol{s}_{t-1}) \right)$$

$$\log \frac{p_{T_0}(\boldsymbol{s}_0 = \boldsymbol{x}|\boldsymbol{s}_1) \left( \prod_{t=2}^K p_{T_t}(\boldsymbol{s}_{t-1}|\boldsymbol{s}_t) \right) p^*(\boldsymbol{s}_K)}{q_{T_0}\boldsymbol{x} q_{T_1}(\boldsymbol{s}_1|\boldsymbol{s}_0 = \boldsymbol{x}) \left( \prod_{t=2}^K q_{T_t}(\boldsymbol{s}_t|\boldsymbol{s}_{t-1}) \right)} d\boldsymbol{s}_1^K. \quad (8)$$

This bound is valid for any $q$ but will be tight when $q(s_K, s_{K-1}, \ldots, s_1|s_0) = p(s_K, s_{K-1}, \ldots, s_1|s_0)$, and otherwise can be used to obtain a variational training objective. Note that both $q$ and $p$ can be decomposed as a product of one-step conditionals. Here, we can make most of the $q_{T_t}$ transition probabilities match their corresponding $p_{T_t}$ transition probabilities exactly, i.e., for $1 \leq t < K$ we use

$$q_{T_t}(\boldsymbol{s}|\boldsymbol{s}') = p_{T_t}(\boldsymbol{s}|\boldsymbol{s}'). \quad (9)$$

The only approximations will be on both ends of the sequence:

- Sampling exactly from the model's $p(v = x)$ is typically not feasible exactly (it involves the usual posterior inference, e.g., as used in VAEs) but as explained below we will exploit properties of the algorithm to approximate this efficiently. We call the chosen approximation $q_1(v)$.
- At the last step, the optimal $q_{T_K}(\boldsymbol{s}_K|\boldsymbol{s}_{K-1})$ is not simply the model's transition operator at temperature $T_K$, because this conditional also involves the marginal "starting distribution" $p^*(\boldsymbol{s}_K)$. However, because we have picked $T_K$ large enough to make samples from $q_{T_{max}}(\boldsymbol{s}_K|\boldsymbol{s}_{K-1})$ dominated by noise of the same variance as that of $p^*$, we expect the approximation to be good too.

## 3.1 ESTIMATING THE LOG-LIKELIHOOD USING IMPORTANCE SAMPLING

In practice we cannot compute $\mathcal{L}$ exactly (nor its gradient), but we can easily obtain an unbiased estimator of $\mathcal{L}$ (or of its gradient) by sampling $\boldsymbol{s}_1^K$ from the $q$ distributions, i.e., approximate the $\mathcal{L}$ integral by a single Monte-Carlo sample. This is what is done by the training procedure outlined in Algorithm 1, which thus performs stochastic gradient ascent on the variational bound $\mathcal{L}$, and this will

tend to also push up the log-likelihood $\log p(\boldsymbol{x})$ of training examples $\boldsymbol{x}$. Note that such variational bounds have been used successfully in many learning algorithms in the past (Kingma & Welling, 2013; Lamb et al., 2016).

We derive an estimate of the negative log-likelihood by the following procedure. For each training example $x$, we sample a large number of diffusion paths. We then use the following formulation to estimate the negative log-likelihood.

$$\log p(\boldsymbol{x}) = \log \mathbb{E}_{\boldsymbol{x} \sim p_{\mathcal{D}}, q_{T_0}(\boldsymbol{x}) q_{T_1}(\boldsymbol{s}_1 | \boldsymbol{s}_0(\boldsymbol{x},)) \left( \prod_{t=2}^{K} q_{T_t}(\boldsymbol{s}_t | \boldsymbol{s}_{t-1}) \right)}$$

$$\left[ \frac{p_{T_0}(\boldsymbol{s}_0 = \boldsymbol{x} | \boldsymbol{s}_1) \left( \prod_{t=2}^{K} p_{T_t}(\boldsymbol{s}_{t-1} | \boldsymbol{s}_t) \right) p^*(\boldsymbol{s}_K)}{q_{T_0}(\boldsymbol{x}) q_{T_1}(\boldsymbol{s}_1 | \boldsymbol{s}_0 = \boldsymbol{x}) \left( \prod_{t=2}^{K} q_{T_t}(\boldsymbol{s}_t | \boldsymbol{s}_{t-1}) \right)} \right] \tag{10}$$

## 4 TRANSITION OPERATORS FOR VARIATIONAL WALKBACK

Up to now we have not specified what the form of the transition operators should be. Two main variants are possible here. Either we directly parametrize the transition operator, like with denoising auto-encoders or generative stochastic networks, or we obtain our transition operator implicitly from some energy function, for example by applying some form of Gibbs sampling or Langevin MCMC to derive a transition operator associated with the energy function.

An advantage of the direct parametrization is that it eliminates the constraint to have symmetric weights, which is interesting from the point of view of biological plausibility of such algorithms. An advantage of the energy-based parametrization is that at the end of the day we get an energy function which could be used to compute the unnormalized joint probability of visible and latent variables. However, note that in both cases we can easily get an estimator of the log-likelihood by simply using our lower bound $\mathcal{L}$, possibly improved by doing more expensive inference for $p_{T_K}(\boldsymbol{s}_K | \boldsymbol{s}_{K-1})$.

### 4.1 PARAMETRIC TRANSITION OPERATOR

In our experiments we considered Bernoulli and isotropic Gaussian transition operators for binary and real-valued data respectively.

When we sample from the transition operator we do not attempt to pass gradients through the sampling operation. Accordingly, backpropagation is performed locally on each step of the walk-back, and there is no flow of gradient between multiple walk-back steps.

Additionally, we use a "conservative" transition operator that averages its input image together with the sample from the learned distribution (or takes a weighted average with a fixed $\alpha$ weighting) for the transition operator. Just after parameter initialization, the distribution learned by the transition operator's output is essentially random, so it is very difficult for the network to learn to reconstruct the value at the previous step.

**Bernoulli Transition Operator**

$$\rho = sigmoid\left( \frac{(1-\alpha) * x_{t-1} + \alpha * F_\rho(x_{t-1})}{T_t} \right) \tag{11}$$

**Gaussian Transition Operator**

$$\mu = (1-\alpha) * x_{t-1} + \alpha * F_\mu(x_{t-1}) \tag{12}$$

$$\sigma = sigmoid(T_t \log(1 + e^{F_\sigma(x_{t-1})})) \tag{13}$$

$F_\rho, F_\mu, F_\sigma$ are functions (in our case neural networks) which take the previous x value from the walkback chain and return estimates of the value of $\mu$ and $\sigma$ respectively. $T$ is the temperature which is dependent on the walkback step $t$. $x_{t-1}$ is the previous value in the walkback chain.

## 5 RELATED WORK

**Contrastive Divergence**

This algorithm is clearly related to the contrastive divergence algorithm with $k = T$ steps (CD-$k$). The CD-$k$ algorithm approximates the log-likelihood gradient by trying to match the sufficient statistics with the data clamped to the sufficient statistics after $k$ steps of the transition operator. The parameter update is the difference of these sufficient statistics, which also corresponds to pushing down the energy of the data-clamped configuration while pushing up the energy of the random variables after $k$ steps of the transition operator.

Two important differences are that, because the temperature is increasing in the variational walk-back procedure,

1. the energy gradients $\frac{\partial E(\boldsymbol{s})}{\partial \boldsymbol{s}}$ do not cancel each other telescopically along the chain from $\boldsymbol{s}_0$ to $\boldsymbol{s}_T$,

2. as $t$ increases we move more and more randomly rather than following the energy of the model, allowing to hunt more effectively the areas near spurious modes.

A third difference is that the learning procedure is expressed in terms of the transition operator rather than directly in terms of the energy function. This allows one to thus train a transition operator directly, rather than indirectly via an energy function.

**Generative Stochastic Networks**

The Generative Stochastic Networks (GSN) algorithm proposed by Bengio et al. (2013b) learns a transition operator by iteratively injecting noise and minimizing the reconstruction error after a number of transition operator steps starting at a data point, and back-propagating through all these steps. One thing in common is the idea of using the walkback intuition instead of isotropic noise in order to converge more efficiently. A major difference is that the algorithm proposed for GSNs involves the minimization of overall reconstruction error (from the input data point $x$ to the sampled reconstruction many steps later). This will tend to blur the learned distribution. Instead, the variational walk-back algorithm minimizes reconstruction error one step at a time along the walk-away trajectory.

In addition, the variational walkback GSNs require back-propagating through all the iterated steps, like the DRAW algorithm (Gregor et al., 2015). Instead the variational walk-back algorithm only requires back-propagating through a *single step* at a time of the transition operator. This should make it easier to train because we avoid having to optimize a highly non-linear transformation obtained by the composition of many transition operator steps.

**Non-Equilibrium Thermodynamics**

There are two main differences between the Variational Walkback algorithm and the Non-Equilibrium Thermodynamics:

1. Instead of isotropic noise to move away from the data manifold, we propose to use the model's own transition operator, with the idea that it will "seek and destroy" the spurious modes much more efficiently than random moves.

2. Instead of injecting a fixed amount of noise per time step, we increase the noise as it moves away from the data manifold, and anneal the noise when we are close to the data manifold. This way, we can quickly reach the noise prior without loosing the details of the data. Our model takes significantly fewer steps to walk away and back to the manifold, as compared to the 1000 steps used for Non-Equilibrium Thermodynamics.

**Annealed Importance Sampling (AIS)**

Annealed Importance Sampling is a sampling procedure. Like variational walkback, it uses an annealing schedule corresponding to a range of temperature from infinity to 1. It is used to estimate a partition function. Unlike Annealed Importance Sampling, variational walkback is meant to provide a good variational lower bound for training a transition operator.

**Reverse Annealed Importance Sampling Estimator (RAISE)**

RAISE is a reverse AIS, as it starts from a data point and then increases the temperature. In this way it is similar to the Q-chain in variational walkback. The advantage of RAISE over AIS is that it yields an estimator of the log-likelihood that tends to be pessimistic rather than optimistic, which makes it better as an evaluation criteria.

Like AIS, RAISE estimates the log-likelihood using a form of importance sampling, based on a product (over the chain) of the ratios of consecutive probabilities (not conditional probabilities from the model). Variational walkback does not work with estimates of the model's unconditional probability, and instead works directly with a conditional probability defined by the transition operator. It is for this reason that variational walkback does not need to have an explicit energy function).

## 6 EXPERIMENTS

We evaluated the variational walkback on three datasets: MNIST, CIFAR (Krizhevsky & Hinton, 2009), and CelebA (Liu et al., 2015). The MNIST and CIFAR datasets were used as is, but the aligned and cropped version of the CelebA dataset was scaled from 218 x 178 pixels to 78 x 64 pixels and center-cropped at 64 x 64 pixels (Liu et al., 2015). For all of our experiments we used the Adam optimizer (Kingma & Ba, 2014) and the Theano framework (Al-Rfou et al., 2016). The training procedure and architecture are detailed in appendix A.

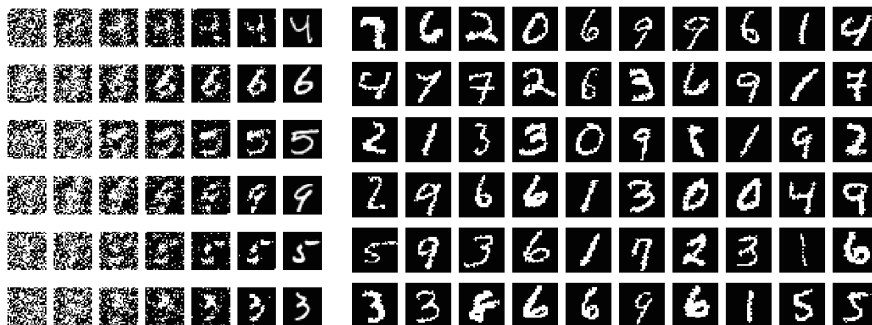

Figure 1: Samples on MNIST using a Bernoulli likelihood in the transition operator, 8 walkback steps during training, and 13 walkback steps during sampling. On right. Diffusion process for sampling MNIST digits starting from bernoiulli noise. This shows how the variational walkback iteratively generates images starting from a noise prior. For intermediate steps we display samples and for the final step (right) we display the transition operator's mean.

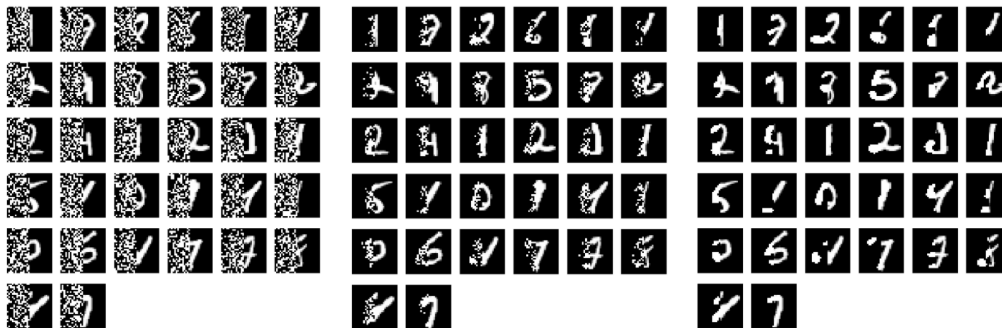

Figure 2: Variational Walkback Inpainting MNIST the left half of digits conditioned on the right half. The goal is to fill in the left half of an MNIST digit given an observed right half of an image (drawn from validation set).

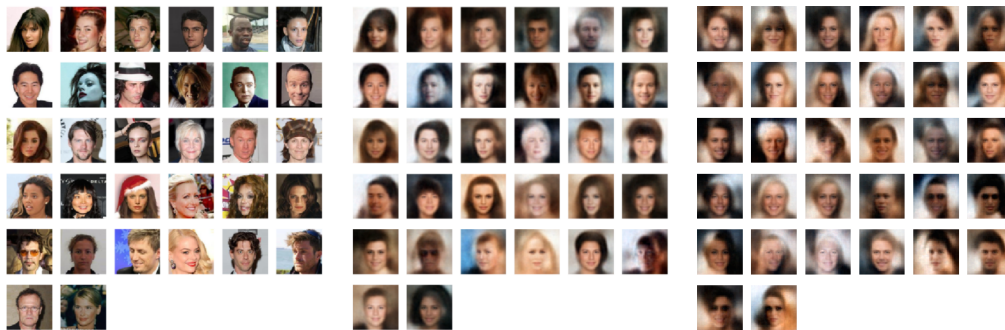

Figure 3: Original Images from CelebA (left), Variational Walkback Reconstructions (middle) and Samples (right).

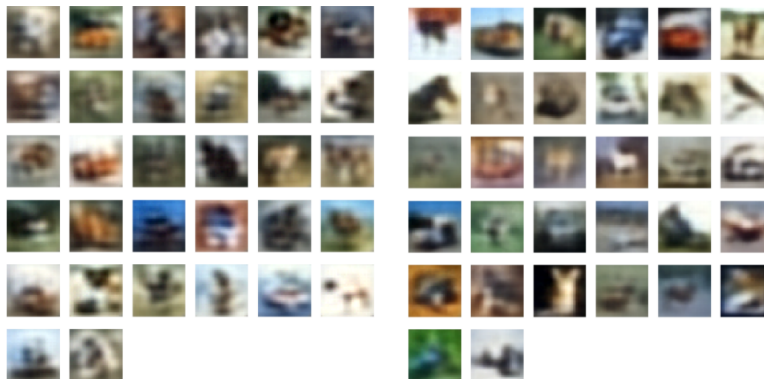

Figure 4: Variational Walkback Samples on CIFAR10 (left and right).

We reported samples on CIFAR, MNIST, CelebA and inpainting results on MNIST. Our inpainting results on MNIST are competitive with generative stochastic networks and show somewhat higher consistency between the given part of the image and the generated portion (Bengio et al., 2013c). However, we note that our samples on CIFAR and CelebA show the same "blurring effect" that has been observed with autoencoder-based generative models trained to minimize reconstruction loss (Lamb et al., 2016).

## 7 CONCLUSION AND FUTURE WORK

We have introduced a new form of walk-back and a new algorithm for learning transition operators or undirected graphical models. Our algorithm learns a transition operator by allowing the model to walk-away from the data towards the noise prior and then teaching it to actually to have its transitions trained to go backwards each of these walk-away steps, i.e., towards the data manifold. Variational walk-back increases the temperature along the chain as it is moving further away from the data manifold, and inversely, anneals the temperature at generation time, as it gets closer to the estimated manifold. This allows the training procedure to quickly find and remove dominant spurious modes. Learning a transition operator also allows our model to learn only a conditional distribution at each step. This is much easier to learn, since it only needs to capture a few modes per step. The model also only locally carves the energy function, which means that it does not have to learn the entire joint probability distribution, but rather steps towards the right direction, making sure that everywhere it puts probability mass as well as around the data, the energy gradient is pointing towards the data.

Our experimental results have confirmed that the model can walk towards the data manifold in a few steps, even when the modes are sharp.

Future work should extend this algorithm and experiments in order to incorporate latent variables. The state would now include both the visible $\vec{x}$ and some latent $\vec{h}$. Essentially the same procedure can be run, except for the need to initialize the chain with a state $\vec{s} = (\vec{x}, \vec{h})$ where $\vec{h}$ would ideally be an estimate of the posterior distribution of $\vec{h}$ given the observed data point $\vec{x}$. Another interesting direction to expand this work is to replace the log-likelihood objective at each step by a GAN-like objective, thus avoiding the need to inject noise independently on each of the pixels, during one application of the transition operator, and allowing the latent variable sampling to inject all the required high-level decisions associated with the transition. Based on the earlier results from Bengio et al. (2013a), sampling in the latent space rather than in the pixel space should allow for better generative models and even better mixing between modes.Bengio et al. (2013b)

## ACKNOWLEDGMENTS

The authors would like to thank Benjamin Scellier and Aaron Courville for their helpful feedback and discussions, as well as NSERC, CIFAR, Google, Samsung, Nuance, IBM and Canada Research Chairs for funding, and Compute Canada for computing resources.

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

## A    ARCHITECTURE DETAILS

The architecture that was used for the CelebA and CIFAR dataset was similar to the architecture used by Lamb et al. (2016), with a convolutional encoder followed by two fully connected hidden layers, followed by a decoder with strided convolutions (Radford et al., 2015). Batch norm was applied in all layers except for the last layer. For all layers except for the last we used the *tanh* activation function. Surprisingly, we were unable to obtain good results using the RELU or Leaky RELU activation .

On the binarized MNIST dataset we used a transition operator with Bernoulli outputs. A feed-forward neural network was used to estimate the parameters (per-pixel probabilities) for the Bernoulli outputs. This neural network consisted of a single hidden layer with 4096 hidden units and the tanh activation function.

## B    WALKBACK PROCEDURE DETAILS

The variational walkback algorithm has three unique hyperparameters. One is the number of walk-back steps performed during training. Another is the number of walkback steps performed when sampling from the model. Still another is the temperature schedule used during training, reconstruction, or sampling.

The most conservative hyperparameter setting would involve using a large number of walkback steps during training and slowly increasing the temperature. However, this could make training slow, and if too few steps are used, the end of the walkback chain will not match the noise prior, leading to low quality samples.

A dynamic approach to setting the number of walkback steps and temperature schedule may be possible, but in this work we set these hyperparameters empirically. We found that during training using a temperature schedule of $T = T_0\sqrt{2^t}$ produced good results, where $T_0 = 1.0$ is the initial temperature and $t$ is the step index. During sampling, we found good results using the reverse schedule: $T = \frac{\sqrt{2^N}}{\sqrt{2^t}}$, where $t$ is the step index and $N$ is the total number of sampling steps.

For MNIST, we achieved our results using 8 training steps of walkback. For CIFAR, we used 15 training steps and 20 sampling steps. For CelebA, we used 30 training steps and 35 sampling steps. In general, we found that we could achieve higher quality results by using more steps during sampling then we used during training. We found that more difficult datasets, like CIFAR and CelebA, required longer walkback chains. Finally, our model is able to achieve results competitive with Non-Equilibrium Thermodynamics (Sohl-Dickstein et al., 2015), despite that method requiring chains with far more steps (1000 steps for MNIST).

## C    ALTERNATIVE FORMULATION OF VARIATIONAL BOUND

The marginal probability of $\boldsymbol{v} = \boldsymbol{x}$ at the end of the above $K$-step process is thus

$$p(\boldsymbol{x}) = \int_{\boldsymbol{s}_1^K} \left( \prod_{t=1}^{K} p_{T_t}(\boldsymbol{s}_{t-1}|\boldsymbol{s}_t) \right) p^*(\boldsymbol{s}_K) d\boldsymbol{s}_1^K \tag{14}$$

where $T_t$ is an annealing schedule with $T_0 = 1$ and $T_K = T_{\max}$ and $p^*$ is the "starting distribution", such as the Gaussian of variance $\sigma_{\text{data}}^2$. We can rewrite this as follows by taking the log and multiplying and dividing by an arbitrary distribution $q(\boldsymbol{s}_1, \ldots, \boldsymbol{s}_K)$ decomposed into conditionals $q_{T_t}(\boldsymbol{s}_t|\boldsymbol{s}_{t-1})$:

$$q(\boldsymbol{s}_0, \boldsymbol{s}_1, ..., \boldsymbol{s}_k) = \left( \prod_{t=1}^{K} q_{T_t}(\boldsymbol{s}_t|\boldsymbol{s}_{t-1}) \right) q(\boldsymbol{s}_K) \tag{15}$$

giving us:

$$\log p(\boldsymbol{x}) = \log \int_{\boldsymbol{s}_1^K} q_{T_0}(\boldsymbol{x}) \left( \prod_{t=1}^{K} q_{T_t}(\boldsymbol{s}_t|\boldsymbol{s}_{t-1}) \right) \frac{\left( \prod_{t=1}^{K} p_{T_t}(\boldsymbol{s}_{t-1}|\boldsymbol{s}_t) \right) p^*(\boldsymbol{s}_K)}{q_{T_0}(\boldsymbol{x}) \left( \prod_{t=1}^{K} q_{T_t}(\boldsymbol{s}_t|\boldsymbol{s}_{t-1}) \right)} d\boldsymbol{s}_1^K \qquad (16)$$

where we understand that $\boldsymbol{s}_0 = \boldsymbol{x}$. Now we can apply Jensen's inequality as usual to obtain the variational bound

$$\log p(x) \geq \mathcal{L} \quad = \int_{\boldsymbol{s}_1^K} q_{T_0}(\boldsymbol{x}) \left( \prod_{t=1}^{K} q_{T_t}(\boldsymbol{s}_t|\boldsymbol{s}_{t-1}) \right) \log \frac{\left( \prod_{t=1}^{K} p_{T_t}(\boldsymbol{s}_{t-1}|\boldsymbol{s}_t) \right) p^*(\boldsymbol{s}_K)}{q_{T_0}(\boldsymbol{x}) \left( \prod_{t=1}^{K} q_{T_t}(\boldsymbol{s}_t|\boldsymbol{s}_{t-1}) \right)} d\boldsymbol{s}_1^K. \quad (17)$$

## D   Tightness of the Variational Bound

We present an argument that running the walkback chain for a sufficient number of steps will cause the variational bound to become tight.

Consider a sequence $s_t, ..., s_1$ generated in that order by our model p through a sequence of applications of the transition operator T, i.e., $p(s_1, ..., s_t) = p(s_t)T(s_{t-1}|s_t)...T(s_1|s_2)$, i.e. $p(s_{n-1}|s_n) = T(s_{n-1}|s_n)$, but note that $p(s_n|s_{n-1}) \neq p(s_{n-1}|s_n)$.

Let $p_i(s)$ denote the stationary distribution associated with T. Note that T and $p_i$ and related by the detailed balance equation, i.e., $T(s|s')p_i(s') = T(s'|s)p_i(s)$.

We want to approximate the posterior

$p(s_t, s_{t-1}, ..., s_2|s_1) = \prod_{n=2}^{t} p(s_n|s_{n-1})$

now by Bayes rule

$= \prod_{n=2}^{t} p(s_{n-1}|s_n) \frac{p(s_n)}{p(s_{n-1})}$ by telescopic cancellation and definition of T

$= \frac{p(s_t)}{p(s_1)} \prod_{n=2}^{t} T(s_{n-1}|s_n)$ now by detailed balance

$= \frac{p(s_t)}{p(s_1)} \prod_{n=2}^{t} T(s_n|s_{n-1}) \frac{p_i(s_{n-1})}{p_i(s_n)}$ by telescopic cancellation

$= \frac{p(s_t)}{p_i(s_t)} \frac{p_i(s_1)}{p(s_1)} \prod_{n=2}^{t} T(s_n|s_{n-1})$ again by definition of T

$= \frac{p(s_t)}{p_i(s_t)} \frac{p_i(s_1)}{p(s_1)} \prod_{n=2}^{t} p(s_n|s_{n-1})$

So our approximation error in the posterior is the factor $\frac{p(s_t)}{p_i(s_t)} \frac{p_i(s_1)}{p(s_1)}$.

If t is large enough, then $s_1$ (being at the end of the generative sequence) has pretty much converged, i.e., $p(s_1) \approx p_i(s_1)$.

If we throw in temperature annealing along the way (now the notation would have to be changed to put an index n on both p and T), with the initial temperature being very high, then we can hope that the initial Gaussian $p(s_t)$ is very similar to the stationary distribution at high temperature $p_i(s_t)$.

These arguments suggest that as we make t larger and the final (initial) temperature larger as well, the approximation becomes better.

