# Peer review of "The Variational Walkback Algorithm"

_ICLR 2017 — rejected_

[Official Review · AnonReviewer1 · rating 5 · confidence 5 · 16 Dec 2016]
**Like the underlying idea, not convinced by its current incarnation**

I very much like the underlying idea for this paper. I wasn't convinced by the execution in its current state. My primary concern is the one I expressed in my pre-review question below, which I don't think the authors addressed. Specifically, I think the choice of q(s | s') = p(s | s') will make the forward and reverse trajectories almost pathologically mismatched to each other, and will thus make the variational bound extremely loose and high variance. 

The claim about the tightness of the bound in Appendix D relies on the assumption that the transition distribution obeys detailed balance. The learned transition distribution in the paper does not obey detailed balance, and therefore the tightness claim in Appendix D does not hold. (In Section 2.1 you briefly discuss the idea of learning an energy function, rather than directly learning a transition distribution. I think this would be excellent, and in that case you could choose an MCMC transition operator that does obey detailed balance for that energy function.) I did not go through Appendix D beyond this step.

The experimental results were not visually impressive. I suspect this is primarily driven by the mismatch between generative and inference trajectories. See my concern above and in the pre-review question below.

Also, see note below for sec. 5. I suspect some terms are being dropped from the training gradient.

The paper is optimizing a variational bound on log likelihood. You should really, really, really report and compare log likelihoods against competing methods!

Detailed comments below. Some of these were written based on a previous version of the paper.
sec 1.2 - first paragraph is very difficult to follow
"these modes these spurious modes" -> "these spurious modes"
sec 2.1 - "s = (v,h)" -> "s = {v,h}"
sec 2.2 - "with an MCMC" -> "with an MCMC chain"
"(ideally an MCMC)" -> "(e.g. via MCMC)" MCMC is not ideal ... it's just often the best we can do.
sec 3, last bullet - could make the temperature infinite for the last step, in which case the last step will sample directly from the prior, and the posterior and the prior will be exactly the same.
sec. 4 -- Using an energy function would be great!! Especially, because many MCMC transition operators obey detailed balance, you would be far less prone to suffer from the forward/backward transition mismatch that is my primary concern about this technique.
eq. 12,13 -- What is alpha? How does it depend on the temperature. It's never specified.
sec. 5, last paragraph in GSN section -- Note that q also depends on theta, so by not backpropagating through the full q chain you are dropping terms from the gradient.
sec. 5, non-equilibrium thermodynamics -- Note that the noneq. paper also increases the noise variance as the distance from the data increases.
Fig. 1 -- right/left mislabeled
Fig. 2 -- label panes
Fig. 3 -- After how many walkback steps?

[Official Review · AnonReviewer3 · rating 4 · confidence 5 · 16 Dec 2016]
**clever idea, but needs more quantitative validation and discussion of (closely) related work**

This paper proposes a new kind of generative model based on an annealing process, where the transition probabilities are learned directly to maximize a variational lower bound on the log-likelihood. Overall, the idea is clever and appealing, but I think the paper needs more quantitative validation and better discussion of the relationship with prior work.

In terms of prior work, AIS and RAISE are both closely related algorithms, and share much of the mathematical structure with the proposed method. For this reason, it’s not sufficient to mention them in passing in the related work section; those methods and their relationship to variational walkback need to be discussed in detail. If I understand correctly, the proposed method is essentially an extension of RAISE where the transition probabilities are learned rather than fixed based on an existing MRF. I think this is an interesting and worthwhile extension, but the relationship to existing work needs to be clarified.

The analysis of Appendix D seems incorrect. It derives a formula for the ratios of prior and posterior probabilities, but this formula only holds under the assumption of constant temperature (in which case the ratio is very large). When the temperature is varied, the analysis of Neal (2001) applies, and the answer is different. 

One of the main selling points of the method is that it optimizes a variational lower bound on the log-likelihood; even more accurate estimates can be obtained using importance sampling. It ought to be easy to report log-likelihood estimates for this method, so I wonder why such estimates aren’t reported. There are lots of prior results to compare against on MNIST. (In addition, a natural baseline would be RAISE, so that one can check if the ability to learn the transitions actually helps.)

I think the basic idea here is a sound one, so I would be willing to raise my score if the above issues are addressed in a revised version.


Minor comments:

“A recognized obstacle to training undirected graphical models… is that ML training requires sampling from MCMC chains in the inner loop of training, for each example.” This seems like an unfair characterization, since the standard algorithm is PCD, which usually takes only a single step per mini-batch.

Some of the methods discussed in the related work are missing citations.

The method is justified in terms of “carving the energy function in the right direction at each point”, but I’m not sure this is actually what’s happening. Isn’t the point of the method that it can optimize a lower bound on the log-likelihood, and therefore learn a globally correct allocation of probability mass?

[Official Review · AnonReviewer4 · rating 4 · confidence 4 · 20 Dec 2016]

The authors present a method for training probabilistic models by maximizing a stochastic variational-lower-bound-type objective. Training involves sampling and then learning a transition-based inference to "walk back" samples to the data. Because of its focus on transitions, it can be used to learn a raw transition operator rather than purely learning an energy-based model. The objective is intuitively appealing because of its similarity to previous successful but less principled training methods for MRFs like Contrastive Divergence.

The idea for the algorithm is appealing, and it looks like it could find a nice place in the literature. However, the submission in its current form is not yet ready for publication. Experiments are qualitative and the generated samples are not obviously indicative of a high model quality. As pointed out elsewhere, the mathematical analysis does not currently demonstrate tightness of the variational bound in the case of a learned transition operator. More evaluation using e.g. annealed importance sampling to estimate held-out likelihoods is necessary. Assuming that the analysis can be repaired, the ability to directly parametrize a transition operator, an interesting strength of this method, should be explored in further experiments and contrasted with the more standard energy-based modeling.

This looks like a promising idea, and other reviews and questions have already raised some important technical points which should help strengthen this paper for future submission.

[Final Decision · Program Chairs · 06 Feb 2017]
**ICLR committee final decision**

This paper proposes an algorithm for training undirected probabilistic graphical models. However, there are technical concerns of correctness that haven't been responded to. It also wasn't felt the method was evaluated appropriately.